# Relationship between Helicopter Parenting and Chinese Elementary School Child Procrastination: A Mediated Moderation Model

**DOI:** 10.3390/ijerph192214892

**Published:** 2022-11-12

**Authors:** Ronghua Zhang, Huanrong Zhang, Xiaofeng Guo, Jiali Wang, Zhongxiang Zhao, Lean Feng

**Affiliations:** 1School of Psychology, Northwest Normal University, Lanzhou 730071, China; 2Lanzhou Qilihe Elementary School, Lanzhou 730050, China; 3Gansu Academy of Social Science, Lanzhou 730070, China

**Keywords:** over-parenting, child procrastination, child self-control, parental smartphone use

## Abstract

Background: The family environment is essential for elementary school children’s development. With smartphone penetration into all aspects of people’s lives, how parenting affects children’s behavior may show new patterns. Objective: This study constructed a mediated moderation model, focusing on the mediating role of child self-control and parental phubbing to clarify the relationship between helicopter parenting (over-parenting) and child procrastination and its mechanisms. Methods: The Smartphone Addiction Scale for Chinese Adults, Brief Self-Control Scale, Over-Parenting Questionnaire, and Short General Procrastination Scale were employed to investigate 562 elementary school-age children and their parents. Results: After data analysis, this study showed the following: (1) helicopter parenting was significantly and positively related to child self-control, child procrastination, and parental smartphone use; (2) child self-control partially mediated the relationship between helicopter parenting and child procrastination; and (3) pathways between helicopter parenting and child self-control were moderated by mother-phubbing behavior. Conclusion: These findings inform parents of their roles in family education.

## 1. Introduction

As an old Chinese poem known to almost everyone, the Song of Tomorrow said: 

“Tomorrow and tomorrow again; how many tomorrows then? If we always wait for another day in vain, our lives will pass away. Make no delay in doing anything, or you will grow old when autumn comes after spring!”

This wisdom stems from 600 years ago, trying to wake people up from procrastination. Even now, human beings, especially children, still cannot eliminate procrastination. Data has shown nearly 15%–20% of adults have chronic procrastination [1], and it is more prevalent among younger adults, as 70% of students admit that they procrastinated, especially in the academic field [2]. Procrastination is a self-regulation failure that leads to poor performance and reduced well-being [3]. Although procrastination may have positive effects, such as avoiding stress [4], most research considers the negative side. For elementary students, procrastination is always correlated with poor academic performance, negative emotions, such as anxiety, self-negation, and harm to mental and physical health development [5,6]. Previous research found several factors determine procrastination, for instance, internal factors such as individual neurodevelopment, self-control, or temporal motivation, and external factors such as task-related or environmental affections [7,8]. Among them, family circumstances and self-control may play an essential role in younger children’s procrastination.

In the family education context, wishing children to become successful adults is a common phenomenon among parents. To achieve this, parents may take more actions than their children need, also known as over-parenting or helicopter parenting. Highly authoritarian and controlling “helicopter parents” try to help their children too much. Ironically, not every parent with helicopter parenting can do precisely what their children require. One issue is the use of cell phones. According to an online survey that 11,000 parents participated in, 34% of parents are worried about their children with “cellphone addiction,” and over 84% of parents agreed with the necessity to develop children’s healthy cellphone use habits [9]. With the entry of smartphones into the family and quickly becoming one of the essential pieces of equipment, parents spend more time on their smartphones. Thus, we can always see such a tableau: the word-strict parents ask children to study hard and stay away from smartphones when holding them in their hands. That could make children confuse, they usually observe and learn behavior from their parents, but their parents’ requirements or guidance are controversial with their behaviors [10]. Self-control was another core factor related to younger children’s procrastination; as a critical capacity, self-control showed the same development pattern as procrastination and shared the same neuro-system development [11]. According to the Self-regulation intergenerational transmission model [12], parents’ behavior can affect children’s self-control, which can also predict the child’s procrastination.

Although extensive literature addresses how parenting style affects children’s behavior, few studies have focused on how helicopter parenting and smartphone use influence procrastination in children. The present study employed a model to explore the relationship between helicopter parenting, parental cellphone use, self-control, and children’s procrastination to test how environmental and individual factors affect children’s procrastination.

### 1.1. Helicopter Parenting and Child Procrastination

Procrastination emerges in adults and even elementary school students who postpone their planned tasks. Feng and his colleagues suggested stages of procrastination development based on neural development; specifically, it starts at 6–8, which shows the trend of procrastination, then it develops into procrastinating behavior at 10–12, and finally, becomes the stable procrastination characteristic at 12–15 [13]. Procrastination develops throughout elementary-school ages; thus, the present study focused on the procrastination of elementary school children, as the bio-psycho-social-ecological systems theory suggests that family is one of the essential factors that influence children’s development [14]. Among the family circumstances, how parents rear offspring or parenting style plays a vital role in influencing younger children’s behavior and personal and social well-being [15].

Helicopter parenting, also known as overparenting, refers to a parenting style in which parents are overly concerned and interfere with their children’s lives, making decisions for them, providing them with solutions to problems they might face, and being overprotective [16,17]. Helicopter parents solve children’s issues and prepare to help them at any time [18]. Generally, helicopter parenting is characterized by high support, high control, and low democracy. It is a relatively negative parenting style that runs counter to democratic parenting, is more controlling than authoritative parenting, and is more concerned with limiting children’s behavior than authoritarian parenting [19]. However, most previous research has used the term helicopter parenting for emerging adults or college students, usually from 18–25; in the present study, we borrowed the term for elementary school students. This was mainly because the media paid more attention to how parents hovering over their children made the term more popular among emerging adults [20]. As a parenting method, parents do not start hovering over their children until they grow up. Helicopter parenting started very early throughout the family education span.

Helicopter parenting was assumed to influence children’s procrastination due to specific rearing patterns. According to the self-system processes model, social contexts such as parental involvement and autonomy support can encourage students to gain the basic needs for relatedness, competence, and autonomy [20,21]. When it comes to helicopter parents, the parents showed high control, which lowered their autonomy and made children feel less self-control; high involvement, which raised children’s relatedness and made children dependent on parents when they were in trouble; and high structure, which raised children’s competence, and led children toward perfectionism and fear of loss (Figure 1). For younger children, lower self-control, higher dependence, and higher perfectionism are correlated with procrastination [11,22,23,24]. Additionally, helicopter parenting may cause other negative emotions, such as anxiety, that further induce procrastination in children. Previous research has found that helicopter parenting negatively affects children’s development, deteriorates their mental health, and causes anxiety and depression [25]. According to the short-term mood regulation theory, procrastination is a regulatory strategy to cope with anxiety; when individuals are averse to the task they are facing, they develop negative emotions such as anxiety and resort to procrastination as a way of avoiding the task of regulating this short-term negative mood [26]. Therefore, anxiety generated by helicopter parenting may also contribute to procrastination.

Consequently, it can imply that helicopter parenting can influence children’s procrastination not only in a cognitive but also in an emotional way; thus, we proposed the following hypothesis:

**Hypothesis** **1:***Helicopter parenting positively predicts child procrastination*.

### 1.2. Mediating Role of Children’s Self-Control

Self-control refers to an individual’s ability to change or overcome dominant response tendencies and regulate thoughts, feelings, and behaviors. It is a prerequisite for achieving self-regulation [27]. Many researchers have found that self-control is closely correlated with procrastination [8,11,28,29]. People with lower self-control always prefer short-term benefits to long-term benefits. Children’s failure in self-control is often associated with procrastination; for instance, when children face the choice of doing homework now or playing now, children with low self-control usually would choose the latter and leave work later, which is procrastination. However, previous studies have also found that students procrastinate less by enhancing self-control [30]. Researchers have also found a shared neural structure between self-control and procrastination [8,11,31], implying that self-control may be a factor that influences procrastination. Additionally, several models have explained the mechanism of procrastination, and all regard self-control as an essential factor causing procrastination [8,32,33]. Procee and his colleagues proposed a conceptual model of procrastination, divided into three main group factors that influence procrastination: task-related, personality-related, and other factors such as mood or ego depletion [7]. Self-control was both the individual and status factor and affected how individuals valued the tasks. 

Helicopter parenting affects children’s self-control. Most previous research has regarded helicopter parenting as a negative factor influencing children’s self-control development; for instance, helicopter parenting triggers more risky behaviors in adolescents owing to their reduced levels of self-control [34]. This is mainly because helicopter parents are always ready to help their children and make decisions for their children, which hinders their self-control [35]. Additionally, this highly controlled and low-democracy parenting pattern has also been found to weaken children’s autonomy and reduce their sense of self and self-expression, which finally causes lower self-control [36]. The indirect path whereby parents are dissatisfied with children’s basic psychological needs, such as autonomy, can contribute to children’s lower self-control [37]. However, a few studies have argued that when parents closely monitor their children’s behavior, the decreasing self-control pattern is shallower in children aged 10 to 14 [38]. Specifically, parental control of the youth can help to increase younger children’s self-control [38,39]. Although there are inconsistencies in how helicopter parenting influences children’s self-control, there is no doubt that parenting influences children’s self-control.

Above all, self-control may be a mediating variable in the pathway of helicopter parenting’s effect on procrastination. Therefore, we propose the following hypothesis:

**Hypothesis** **2:***Helicopter parenting predicts child procrastination through the mediating role of children’s self-control*.

### 1.3. Moderating Role of Parental Phubbing Behavior

With the widespread use of smartphones, especially since the worldwide coronavirus disease 2019 pandemic, smartphones have become an essential tool in daily life [40]. In addition to the positive effects of smartphone use during the pandemic, smartphone addiction or overuse has also increased [41]. Phubbing is a blend of “phone” and “snubbing,” which means the interruptions in social interaction caused by one’s mobile phone usage; it is regarded as a kind of social exclusion and interpersonal neglect [42]. In families, parental phubbing behavior is when parents pay excessive attention to their smartphones, resulting in ignoring their children when interacting with them. This has been shown to negatively impact children’s development, mainly affecting parent-child relationships, impairing children’s interpersonal skills, and causing an increase in children’s internalizing problem behaviors [43]. Evidence shows that higher parental phubbing behavior negatively affects children’s self-control. On the one hand, parental phubbing behavior directly influences children’s self-control. According to classical social learning theory [10], children learn their attitudes, values, and behaviors by observing models such as parents. Elementary school children are still learning and trying to develop stable characteristics through the environment. Thus, parental phubbing may cause imitation by children, making them more likely to devote their attention and interest to smartphone use, thus reducing their level of self-control. However, children’s negative emotions, such as anxiety and depression, and the social exclusion caused by parents’ phubbing behaviors may contribute to reduced levels of self-control [44].

Therefore, when parents adopt the “strong pressure, high demand” helicopter parenting style to discipline children, and on the other hand, possess weak self-control and indulge in smartphone use, acting as a phubber (the person who starts phubbing their companion (s)), children feel the contradiction between parents’ words and actions, may consequently develop negative coping styles. Specifically, the path of helicopter parents toward their children’s self-control may be moderated by their parents’ phubbing behavior. Precisely, helicopter parenting may negatively predict children’s self-control. Simultaneously, the influence may be minor if parents show less phubbing; in contrast, if parents show more phubbing behavior, helicopter parenting will have more negative effects on children’s self-control. Thus, we propose the following hypotheses:

**Hypothesis** **3:***Parental phubbing behavior moderates the relationship between helicopter parenting and children’s self-control*.

To test these three hypotheses, we selected 562 elementary school-age children and their parents as study participants to explore the effects of helicopter parenting styles on child procrastination, the mediating role of child self-control in this relationship, and the moderating role of parental phubbing behavior in this relationship (Figure 2).

## 2. Research Methodology

### 2.1. Participants

Convenience sampling was employed, and several first- to sixth-grade students from three elementary schools in Lanzhou City (GDP per capita 73,800 RMB in 2021; the total population was 4.38 million; it was an underdeveloped city located in Northwest China) participated in the study. The elementary schools were recruited from a teacher internship project managed by the university. They were located in three different districts in the city, and all of them volunteered to participate in the survey. We recruited participants from at least one class of different grades in the three schools and ensured each grade was involved. Informed consent forms were signed by the class teacher and the children’s parents. Questionnaires were completed independently by fathers, mothers, and children, and questionnaires completed by fathers and mothers were collected online through the Questionnaire Star platform (www.wjx.com (accessed on 25 November 2021.)). The questionnaires were distributed to the children, and the completed paper and pencil questionnaires were collected. We received 3836 completed questionnaires (*n* = 1035 for fathers, *n* = 1351 for mothers, and *n* = 1450 for children) to obtain a questionnaire including fathers, mothers, and children from the same family. We matched all questionnaires with a unique ID number. The questionnaires that could not be matched to an ID number had unfinished items, had wrong answers in the detective question (e.g., please choose B in this question), or filled all questionnaires with one or regular pattern answers had been rejected. Finally, we obtained 562 valid completed set questionnaires, including father, mother, and children (the demographic information is in Table 1), with a validity rate of 43.95%. This study was approved by the Ethics Committee of the School of Psychology, Northwest Normal University.

### 2.2. Research Tools

In total, four questionnaires were used in the research. Child procrastination, helicopter parenting, and children’s self-control were completed by children through paper-and-pencil tests in the classroom. One trained graduate student as the experimenter helped to explain how to fill in the questionnaire. In contrast, parents’ phubbing behavior was completed separately through a questionnaire APP online. All questionnaires were written in Chinese. The questionnaire for children was pre-tested among several grade 1 elementary school children and asked the teachers for younger grades to ensure that the youngest could understand the questionnaire. The pre-test results showed that all the questionnaires for children were verified to be suitable for elementary school children. 

Child procrastination. The children completed the questionnaire. We used the Short General Procrastination Scale [45]. The Short General Procrastination Scale comprises nine questions and is a unidimensional test. The scale is scored on a 5-point Likert scale, with responses ranging from “not at all” to “always.” All items were simple and easy to understand, even for young children, such as I always say, “do it tomorrow,” or even if it is easy, I usually do not finish it within a few days. Higher scores indicate a more pronounced tendency to procrastinate. In this study, Cronbach’s alpha coefficient for the scale was 0.798.

Helicopter parenting. The children completed a questionnaire. We employed the 5-item “helicopter” parenting scale [19] to assess whether children perceived their parents to be overly involved in their lives. This 5-item scale is unidimensional and is rated on a 5-point Likert scale, ranging from “not at all” to “always,” with higher scores indicating higher levels of helicopter parenting. In this study, Cronbach’s alpha coefficient of the scale was 0.845.

Children’s self-control. To measure the participating children’s self-control, we used the Brief Self-Control Scale [46] to assess children’s level of self-control. The scale comprises seven items in two dimensions: self-discipline and impulse control. A 5-point Likert scale was used for measurement, with responses ranging from “not at all” to “always.” Higher scores indicate higher levels of self-control. In this study, Cronbach’s alpha coefficient for this scale was 0.760 for the children’s population.

Parents’ propensity toward smartphone addiction (phubbing behavior). We used the Smartphone Addiction Scale for Chinese Adults, developed by Chen et al., [47]. The fathers and mothers of the children provided individual responses to the questionnaire. The scale has a broad application scope and is suitable for measuring smartphone addiction among adults. The scale comprises 26 items in six dimensions: app use, app update, withdrawal response, salience, impaired social functioning, and physical discomfort. A 5-point Likert scale was used, with responses ranging from “strongly disagree” to “strongly agree,” with higher scores indicating more severe smartphone addiction. To collect the parents’ phubbing behavior, which can be represented by the frequency of smartphone use and how it impacted social interaction. Thus, we selected the dimensions of withdrawal response, salience, and impaired social functioning for the measurement. In this study, Cronbach’s alpha coefficient for this scale was 0.878 for the fathers’ group and 0.878 for the mothers’ group.

### 2.3. Data Analysis

First, we conducted Harman’s one-factor test to detect common method bias caused by the self-reported questionnaires. After that, the ANOVA and independent *t*-test were applied to evaluate the difference between children’s grades and sex in the children’s procrastination, helicopter parenting, and self-control. To analyze the relationships between each variable, we adopted Pearson’s correlations.

Then, we conducted a mediation model analysis to test the mediating role of children’s self-control during helicopter parenting on children’s procrastination path and the moderating role of parent phubbing. We used SPSS Windows software version 25.0 and the PROCESS macro program by Hayes to analyze the data, including correlation analysis and the bootstrap method [48]. To investigate the mediated moderation model, this study employed Hayes’ PROCESS Macro Models 4 and 7 [48]. Model 4 aims to analyze the mediation model by testing whether child self-control mediates the effect of helicopter parenting on child procrastination with the bootstrapping confidence interval. The PROCESS Model 7 was used for further investigation of the moderation path between helicopter parenting and child self-control. In order to reduce the multicollinearity, all predictor variables were mean-centered. Where the bootstrap method was used, the sample sampling was 5000 times with a confidence interval of 95%. A confidence interval that did not include 0 indicated that the results were statistically significant [48].

## 3. Results

### 3.1. Common Method Bias Detection

After data collection, Harman’s one-factor test was used to assess common method bias. The results of this test showed that there were 10 factors with eigenvalues greater than 1, and the first factor explained 21.029%, which was less than the 40% criterion, indicating that there was no serious common method bias in the data in this study [49].

### 3.2. Means, Standard Deviations, and Correlations of the Variables

The results of the main descriptive statistics of the grade development trend in children’s procrastination, self-control, and helicopter parenting (Figure 3): To detect the development trend of the three variables, we conducted an ANOVA using grade as the independent variable. The results showed a significant effect on children’s self-control, F (5, 556) = 4.133, *p* = 0.001, η2 = 0.036; the post hoc analysis showed that the score of children’s self-control in grades 5 and 6 was significantly higher than that of grades 1 to 4; the children’s procrastination and helicopter parenting were not significant between grades, Fs (5, 556) < 0.419, *p* < 0.836.

To examine the difference between the sex of the variables, we conducted an independent *t*-test between boys and girls on the children’s procrastination, helicopter parenting, self-control, and parental phubbing (Table 2) and found that boys had significantly higher procrastination than girls, and also rated helicopter parenting more than girls. Simultaneously, there was no significant difference in the other variables.

Pearson’s correlation analyses between each variable (Table 3) showed that children’s procrastination, helicopter parenting, self-control, and father and mother phubbing behavior were all significantly correlated. Specifically, children’s self-control was negatively correlated with all other variables, indicating that the higher the number of children with self-control, the less they procrastinated, and the less helicopter parenting, as well as less parental phubbing behavior.

### 3.3. Assessing the Mediating Effect of Child Self-Control on the Relationship between Helicopter Parenting and Child Procrastination

Model 4 of PROCESS and the bootstrap method were used to test the mediating effect. After controlling for children’s gender and grade, we conducted the test with helicopter parenting as the independent variable, child procrastination as the dependent variable, and child self-control as the mediating variable.

The results (Figure 4) showed that the total effect of helicopter parenting on children’s procrastination was significant (β = 0.541, SE = 0.052, CL = [0.4285, 0.6428], *t* = 10.396, *p* < 0.001), while the direct effect of helicopter parenting on children’s procrastination was significant (β = 0.295, SE = 0.046, CL = [0.205, 0.385], *t* = 6.445, *p* < 0.001); it was still significant after the inclusion of the mediating variable children’s self-control (β = −0.697, SE = 0.044, CL = [−0.783, −0.611], *t* = −15.92, *p* < 0.001). The R2 value of the direct model was 0.181, which changed to 0.437 when the model included the mediation variable. The results suggest that child self-control partially mediates the relationship between helicopter parenting and child procrastination.

### 3.4. Moderating Role of Parental Smartphone Use on the Relationships between Helicopter Parenting, Child Self-Control, and Child Procrastination

Model 7 in PROCESS was used to test the moderating effect of parental phubbing behavior using the bootstrap method, with the independent variable being helicopter parenting, the dependent variable being child procrastination, the mediating variable being child self-control, and the moderating variables being father phubbing behavior (Model 1) and mother phubbing behavior (Model 2).

After testing the significant effect of the mediating role of children’s self-control in the model, we tested the moderating role of father and mother phubbing separately. The results (see Table 4) showed that the mediated moderation model of mother phubbing (Model 2) was significant, while the moderating role of paternal phubbing was not significant in the model (Model 1). The regression analysis showed that in Model 2, helicopter parenting negatively predicted children’s self-control (β = −0.333, SE = 0.042, CL = [−0.416, −0.250], *t* = −7.863, *p* < 0.001); mother phubbing can negatively predict children’s self-control (β = −0.067, SE = 0.024, CL = [−0.123, −0.128], *t* = −3.141, *p* < 0.05), and the interaction term of helicopter parenting × mother phubbing also significantly predicted children’s self-control (β = 0.013, SE = 0.005, CL = [0.003, 0.023], *t* = 2.593, *p* = 0.01). The results indicated that mothers phubbed moderate the first half of the mediation model “helicopter parenting → children self-control → children procrastination”.

Further simple slope analysis showed (Figure 5) that helicopter parenting had a negative predictive effect on children’s self-control when the level of mother phubbing was low (M-1SD) (b simple slop = −0.433, *t* = −7.266, *p* < 0.001, CL = [−0.551, −0.316]). Helicopter parenting also had a negative predictive effect on children’s self-control when the level of mother phubbing was high (M+1SD) (b simple slop = −0.232, *t* = −4.204, *p* < 0.001, CL = [−0.340, −0.124]). Compared with low mother phubbing, the high mother phubbing had a relatively small predictive effect, indicating that with the increasing level of mother phubbing, the predictive effect of helicopter parenting tended to have a gradually decreasing predictive impact on children’s self-control.

## 4. Discussion

Parenting is an essential factor influencing elementary school children’s behavior. The present study focused on how parents affected young children in China. This is typically a collectivist culture and advocates parents with high control, high expectations, and intensive support [37]. The fierce competition in society also pushes parents to be strict with their children, even starting in elementary school. Thus, many helicopter parents appeared hovering around their children. Simultaneously, the digital age also breaks into people’s daily lives, and smartphones with numerous dramatic functions have always attracted tired parents. The present study analyzed and found a moderating role for parental phubbing, especially on helicopter parenting and elementary school children’s procrastination.

This study included 562 families. The data showed that children’s self-control increased among elementary school children after fourth grade. Consistent with previous neuroscience research results, 10–12 years of age is a critical period in developing the frontoparietal and limbic systems [50], which are the physical fundamentals of self-control [13]. Moreover, there are two development peaks of self-control before becoming adults. The second peak starts at 10, which is grade 4, which is also the critical point that children develop abstract logical thinking; both factors contribute to the significant development of the children’s self-control.

Another interesting finding was that boys showed significantly higher procrastination and rate helicopter parenting more than girls did. This is consistent with several previous studies on various backgrounds that males procrastinate more than females, especially at a young age [3,51]. This may be because, compared with girls, boys are more willing to take risks and are attracted to new environments [3]. Additionally, boys rated helicopter parenting more than girls did. This may be because in China, boys can be expected to take on more family responsibilities and be treated more strictly by parents, which raises boys using helicopter parenting more than raising girls objectively.

### 4.1. Mediating Role of Child Self-Control on the Relationship between Helicopter Parenting and Child Procrastination

The study findings validate Hypotheses 1 and 2 since helicopter parenting significantly predicted child procrastination, and child self-control partially mediated this pathway. Thus, it can be concluded that helicopter parenting causes procrastination by reducing children’s self-control levels.

Consistent with previous studies, in this study, helicopter parenting predicted children’s procrastination [52]. Helicopter parenting is a highly controlling parenting style that exerts tremendous behavioral and psychological constraints on children. School-aged children are in a period of self-development, including developing self-esteem and self-evaluation [53]. Helicopter parenting significantly reduces children’s level of self-control, which is in line with self-determination theory and control point theory. Self-determination theory suggests that autonomy, a sense of competence, and relatedness are basic psychological needs, and individuals develop positively when these needs are fulfilled. Conversely, barriers to individuals’ development are created if they remain unfulfilled [54]. Helicopter parenting often leaves children with a reduced sense of autonomy and competence, which leads to less self-control. Furthermore, Rotter’s locus of control theory suggests that externally controlled individuals attribute the outcome of their actions to external factors, such as luck, fate, or the influence of others. In contrast, internally controlled individuals explain the outcome of their actions in terms of their own abilities and efforts. Some studies have found that children are more likely to develop external control beliefs when fathers adopt authoritarian parenting styles [55]. Therefore, children are more likely to reduce their sense of control if they attribute the outcome of their actions to demands from their parents.

### 4.2. Moderating Role of Parental Phubbing Behavior

The study results supported hypothesis H3, showing that mother-phubbing behavior moderates the effects of helicopter parenting on children’s self-control. Unexpectedly, mothers with lower phubbing behavior moderated the path more significantly than mothers with high phubbing behavior. 

It is generally accepted that parental phubbing negatively affects child development [43]. An interesting finding of the present study was the moderating role of mother-phubbing behavior in the impact of helicopter parenting style on children’s self-control. According to the attachment behavior system, mothers play an irreplaceable role in children’s development. It may be inferred that, compared with fathers, mothers’ behavior affected children more.

According to the study results, children’s self-control was lower when mothers used smartphones more often in families with lower helicopter parenting. This may be because, for elementary school children, parental control and demands play an essential role in their development [38]. When parental discipline is higher, it directly contributes to lowering a child’s self-control. In contrast, when parental discipline is lower, children’s self-control is derived more from observing parental behavior, which can be explained well through the social learning theory proposed [10]. Furthermore, in low helicopter parenting conditions, mothers who use smartphones to a lesser extent may communicate more with their children, may be more attentive to children’s emotions and behaviors, have better parent-child relationships, etc. [39], which also benefits children’s development of better self-control. In particular, mothers with higher phubbing behavior decreased children’s self-control more gently than those with a lower phubbing behavior when helicopter parenting increased. Thus, implying that the mother’s low phubbing behavior can be a protective factor only when helicopter parenting is at a low level. 

## 5. Conclusions, Significance, and Limitations of the Study

The main findings of this study were as follows: (1) helicopter parenting positively predicted school-age children’s procrastination; (2) child self-control mediated the relationship between helicopter parenting and child procrastination; and (3) mother phubbing behavior moderated the relationship between helicopter parenting and child self-control.

This study adds knowledge to family education literature and parental behavior styles. First, we found that excessive parental involvement in child development contributed to procrastinating behavior in elementary school children. Early research focused on the positive role of parental involvement in child development and considered parental involvement essential support for child development. However, with advances in research, it is now widely accepted that moderate parental involvement may be better than excessive parental attention [56]. The emergence of buzzwords such as “chicken parenting” demonstrates the educational anxiety of contemporary parents. However, it is more important for family education to provide children with a space to grow independently. Second, the role of parents in family education should not be that of “planners” instead of “participants,” as their behavior greatly influences children. For example, it has been suggested that children’s procrastination is intergenerational, and according to the social learning theory, parents’ procrastination becomes a direct object of imitation for children [57]. In families with a low level of helicopter parenting, if parents exhibit higher smartphone use or low self-control behaviors, then their children’s level of self-control is negatively affected.

The main limitations of the study are as follows: first, the study does not implement longitudinal follow-up to make the findings more explanatory. The results showed that children’s self-control developed during the elementary period and that the development trend was important to scrutinize the path of each variable; the present study used cross-sectional data to describe the relationship between each variable roughly; in the future, longitudinal research can be adopted to make further understanding. Second, the study used self-report questionnaires and required parents to complete them separately online. Because of the lack of supervision, it was not possible to determine whether these questionnaires were completed accurately and honestly; while there were face-to-face guidelines, children may be affected by social desirability. Therefore, dyadic analysis should be adopted in future research, for instance, to induce parents and teachers to rate children’s self-control.

## Figures and Tables

**Figure 1 ijerph-19-14892-f001:**
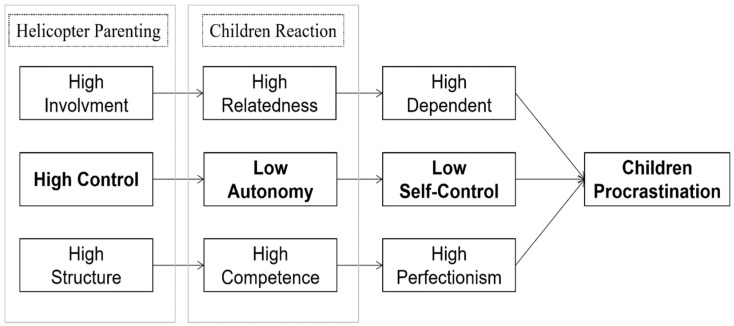
The pathway of helicopter parenting and children’s procrastination based on the Self-System Process Model.

**Figure 2 ijerph-19-14892-f002:**
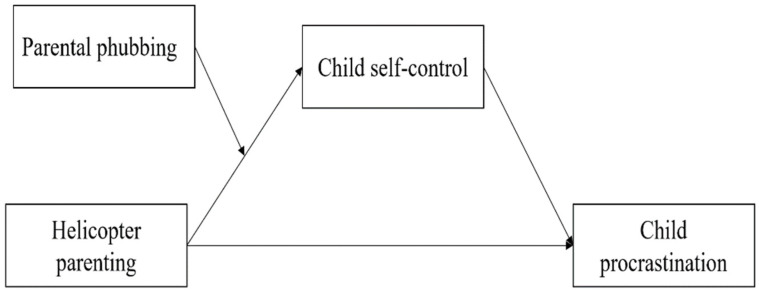
Hypothetical model with moderated mediation effects.

**Figure 3 ijerph-19-14892-f003:**
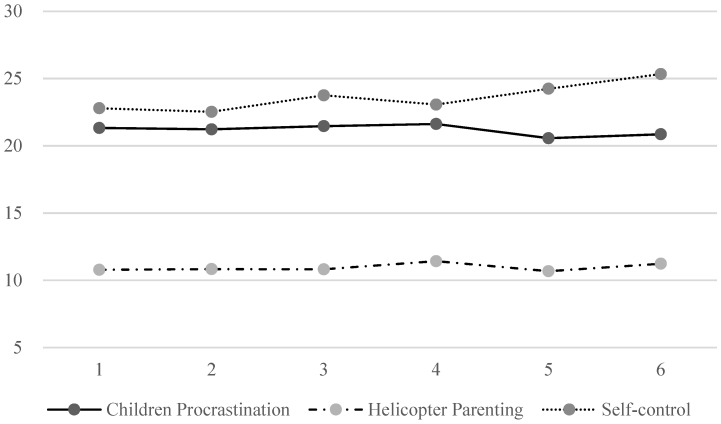
Grade development trend.

**Figure 4 ijerph-19-14892-f004:**
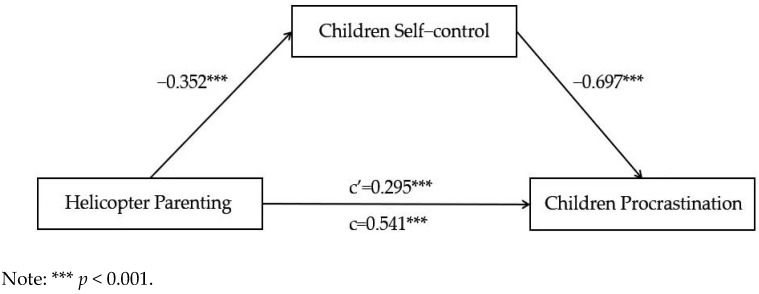
Mediating effects of children’s self-control on the relationship between helicopter parenting and children’s procrastination. C: B = 0.541, SE = 0.052, CL = [0.436, 0.463]; C’: B = 0.295, SE = 0.046, CL = [0.205, 0.385].

**Figure 5 ijerph-19-14892-f005:**
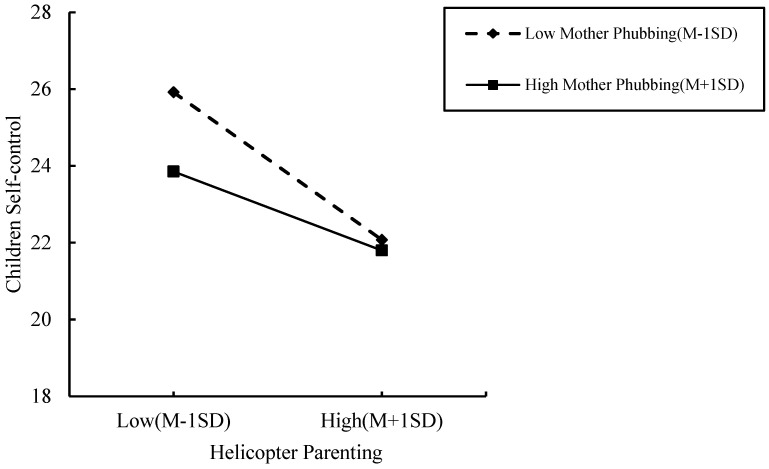
Moderating effects of parental smartphone use on the relationship between helicopter parenting and child self-control.

**Table 1 ijerph-19-14892-t001:** Participants’ Demographic Information.

	Total Sample (*n* = 562)				
	*n*	%	*n*	%	
*Age in Grade*	*Boys*		*Girls*		*Age (M ± SD)*
Grade 1	60	10.7	50	8.9	6.62 ± 0.524
Grade 2	41	7.3	48	8.5	7.51 ± 0.525
Grade 3	62	11	48	8.5	8.53 ± 0.537
Grade 4	49	8.7	57	10.1	9.4 ± 0.491
Grade 5	36	6.4	42	7.5	10.64 ± 0.624
Grade 6	36	6.4	33	5.9	11.51 ± 0.585
*Level of education*	*Father*		*Mother*		
No schooling	9	1.6	18	3.2	
Basic education	57	10.1	65	11.6	
Secondary education	166	29.5	137	24.4	
High School	117	20.8	139	24.8	
Bachelor	204	36.3	193	34.3	
Master/doctorate	9	1.6	10	1.8	
*Level of family income (per year)*					
Less than 12,000 yuan	28	5			
Between 12,000–24,000 yuan	29	5.2			
Between 24,000–48,000 yuan	110	19.6			
Between 48,000–72,000 yuan	170	30.2			
Between 72,000–120,000 yuan	153	27.2			
More than 120,000 yuan	72	12.8			

**Table 2 ijerph-19-14892-t002:** Independent *t*-test between boys and girls.

	Boy (*M* ± *SD)*	Girl (*M* ± *SD)*	*t*	Cohen’s *d*
Children Procrastination	22.10 ± 6.135	20.34 ± 5.707	3.527 ***	0.298
Helicopter Parenting	11.34 ± 4.656	10.57 ± 4.158	2.075 *	0.175
Children Self-control	23.22 ± 4.758	23.79 ± 4.602	−1.442	−0.122
Mother Phubbing	27.92 ± 7.983	28.58 ± 7.428	−1.014	−0.086
Father Phubbing	29.94 ± 7.775	30.64 ± 8.293	−1.017	−0.086

Note: * *p* < 0.05, *** *p* < 0.001.

**Table 3 ijerph-19-14892-t003:** Means, standard deviations, and correlations of the variables.

Item	*M ± SD*	1	2	3	4	5
1. Child procrastination	21.22 ± 5.987	1				
2. Helicopter parenting	10.96 ± 4.430	0.409 **	1			
3. Child self-control	23.50 ± 4.686	−0.615 **	−0.331 **	1		
4. Father phubbing	30.29 ± 8.036	0.253 **	0.223 **	−0.257 **	1	
5. Mother phubbing	28.25 ± 7.714	0.196 **	0.208 **	−0.186 **	0.428 **	1

Note: ** *p* < 0.01.

**Table 4 ijerph-19-14892-t004:** Tests for moderating effects of parental phubbing.

Predict Variables	Child Self-Control (Step 1)			Children Procrastination			Child Self-Control (Step 2)			Child Self-Control (Step 2)		
	*β*	*SE*	*t*	β	*SE*	*t*	*β*	*SE*	*t*	*β*	*SE*	*t*
Helicopter parenting	−0.352	0.042	−8.435 ***	0.295	0.046	6.445 ***	−0.315	0.042	−7.411 ***	−0.333	0.042	−7.863 ***
Child self-control				−0.697	0.044	−15.924 ***						
Father phubbing							−0.116	0.023	−5.057 ***			
Mother phubbing										−0.076	0.024	−3.141 **
Helicopter parenting · Father phubbing							0.008	0.005	1.555			
Helicopter parenting · Mother phubbing										0.013	0.005	2.593 *
*R* ^2^	0.139			0.437			0.180			0.163		
*F*	29.906 ***			108.176 ***			24.373 ***			21.696 ***		

Note: * *p* < 0.05, ** *p* < 0.01, *** *p* < 0.001.

## Data Availability

The data presented in this study are available from the corresponding author. The data are not publicly available due to privacy and ethical considerations.

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
