# Peer review of "Relationship between Helicopter Parenting and Chinese Elementary School Child Procrastination: A Mediated Moderation Model"

_ijerph, 2022, doi:10.3390/ijerph192214892_

Round 1

Reviewer 1 Report

The hypotheses are clearly formulated and well introduced.

The paper uses Process methods as main data processing tools. The authors should present arguments on why these methos were chosen.

The study design does not create a safe environment for children to complete the questionnaire. Even in the pre-test phase children completed questionnaires in the presence of the teacher. This is a serious flaw of the research design as it affects whether children responded honestly on not.

Lines 57-58 read "The controversy between what parents say and what they do may cause children to confuse and result in negative behaviors, such as procrastination". This sentence should be further backed up by evidence and arguments.

Lines 61-62 read: "Family circumstances may affect children’s procrastination through their self-control. " Please explain further how family circumstances impact children's procrastination. How exactly does self-control relate to this?

Lines 102-104 read: "Additionally, negative emotions, particularly anxiety, are another factor that helicopter parenting may influence children’s procrastination." should be rephrased "Additionally, helicopter parenting may cause other negative emotions such as anxiety that further induces procrastination in children". 

Lines 305-306 read "Assessing the Mediating Effect of Child Self-Control on the Relationship Between Helicopter Parenting and Child Procrastination". This is not a sentence.

Lines 378-379 read "On the one hand, boys usually wanted to explore broader spaces, just as our ancestors did; thus, when parents showed the same strength of control, boys felt more constrained than girls subjectively" How does this result from your study or how is it related to the literature?

Reviewer 2 Report

Methodology: Sampling technique needs to be discussed clearly.

Conclusion: It is not clear. The authors need to rewrite this section. Theoritical contribution, practical contribution, limitation of study and future research need to be discussed clearly. 

Reviewer 3 Report

This is an excellently conducted study on very essential issues in child development psychology.

The only thing to note is that the English term phubbing is not generally used and should therefore be understood in the introduction as a contraction of phone and snubbing = phubbing. The meaning is well presented in the introduction.

Round 2

Reviewer 1 Report

The authors addressed all my comments in the previous round of revision. They clarified the methodology especially considering data gathering. The manuscript may be published in the present form.